# Acute Regression in Down Syndrome

**DOI:** 10.3390/brainsci11081109

**Published:** 2021-08-23

**Authors:** Benjamin Handen, Isabel Clare, Charles Laymon, Melissa Petersen, Shahid Zaman, Sid O’Bryant, Davneet Minhas, Dana Tudorascu, Stephanie Brown, Bradley Christian

**Affiliations:** 1Department of Psychiatry, University of Pittsburgh, Pittsburgh, PA 15260, USA; cml14@pitt.edu (C.L.); dam148@pitt.edu (D.M.); dlt30@pitt.edu (D.T.); 2Department of Psychiatry, University of Cambridge, Cambridge CB2 1TN, UK; ichc2@medschl.cam.ac.uk (I.C.); shz10@medschl.cam.ac.uk (S.Z.); sb2403@medschl.cam.ac.uk (S.B.); 3Department of Family Medicine, University of North Texas Health Science Center, Fort Worth, TX 76107, USA; Melissa.Petersen@unthsc.edu (M.P.); SidObryant@unthsc.edu (S.O.); 4Departments of Medical Physics and Psychiatry, University of Wisconsin, Madison, WI 53706, USA; bchristian@wisc.edu

**Keywords:** down syndrome, regression, Alzheimer’s disease, biomarkers

## Abstract

Background: Acute regression has been reported in some individuals with Down syndrome (DS), typically occurring between the teenage years and mid to late 20s. Characterized by sudden, and often unexplained, reductions in language skills, functional living skills and reduced psychomotor activity, some individuals have been incorrectly diagnosed with Alzheimer’s disease (AD). Methods: This paper compares five individuals with DS who previously experienced acute regression with a matched group of 15 unaffected individuals with DS using a set of AD biomarkers. Results: While the sample was too small to conduct statistical analyses, findings suggest there are possible meaningful differences between the groups on proteomics biomarkers (e.g., NfL, total tau). Hippocampal, caudate and putamen volumes were slightly larger in the regression group, the opposite of what was hypothesized. A slightly lower amyloid load was found on the PET scans for the regression group, but no differences were noted on tau PET. Conclusions: Some proteomics biomarker findings suggest that individuals with DS who experience acute regression may be at increased risk for AD at an earlier age in comparison to unaffected adults with DS. However, due to the age of the group (mean 38 years), it may be too early to observe meaningful group differences on image-based biomarkers.

## 1. Introduction

There has been growing interest over the past decade in acute regression among adolescents and young adults with Down syndrome (DS) [1,2]. Characterized by a sudden, and often unexplained, reduction in expressive language, decreased functional living skills and reduced psychomotor activity; regression can result in a significant change in the long-term needs and independence of these individuals. Recent studies have identified a number of potential triggers and associations for regression, including medical issues (e.g., surgery, Hashimoto’s disease, sleep apnea, sleep disruption, menarche and hormonal cycles, depression) and psychosocial stressors (e.g., transition from school, change in living arrangement) [3,4]. However, it is not understood why certain individuals with DS are at risk for regression in response to such events while the vast majority are able to cope effectively. Acute regression is reported to occur in up to 16% of individuals with DS [5] and has also been referred to by a number of other terms, including “new onset autism regression”, “regression, dementia and insomnia”, “catatonia” and “down syndrome degenerative disorder” [6,7]. The disorder appears to occur during adolescence through the mid 20s and can either be sudden onset or progressive. 

Case reports and longitudinal studies of individuals with DS who have experienced acute regression indicate that only about 10% completely regain prior levels of functioning. Approximately 40% improve to some degree, but often fail to regain prior communication skill levels. The remainder fails to regain the skills that were lost [4]. Mircher and colleagues [4] present some of the most recent data on this disorder, describing a cohort of 30 adolescents and young adults with DS who experienced acute regression. In terms of clinical presentation, the most frequently reported psychiatric symptoms included mood disorders (30%), apathy, extreme slowness or catatonia (37%) and stereotypies (27%). Forty percent of the cohort displayed self- or other-directed aggressive behavior. Speech impairment occurred among almost the entire cohort (94%). Structural MRI scans were available for 15 patients and were found to be normal in 11 individuals. The only abnormalities noted were brain atrophy (*n* = 2) and hippocampal abnormalities (*n* = 2). In 2012, Akaloshi et al. [8] described a cohort of 13 adolescents/young adults (mean age 21.2 years) that were diagnosed with acute regression in Japan. All underwent MRI or CT scanning at the time of diagnosis and were treated and followed by the authors. Five of the 13 cases exhibited MRI or CT results that were suggestive of dementia, including mild cerebral white matter ischemia, hippocampal atrophy and basal ganglia calcification. However, similar to the data presented by Mircher et al. [4], no control scans (either of non-affected individuals with DS or neurotypical individuals) were included for comparison. In addition, many of the differences on the MRI and CT scans occur among individuals with DS in the absence of regression, and hippocampal abnormalities, in particular, have been previously documented in the DS population in comparison to neurotypical individuals [9,10]. Following a range of pharmacologic interventions, 23% of the Akaloshi et al. [8] cohort were subsequently rated as “improved”, 54% as “partially improved” and 23% as “no difference”.

Based upon the presentation and loss of skills, some investigators have proposed that regression in DS might be related to dementia [7]. In fact, it is not unusual for individuals with DS who experience acute regression to be given a diagnosis of Alzheimer’s disease (AD) or dementia, based upon the symptom presentation and loss of skills. Consequently, practitioners may prescribe drugs, such as cholinesterase inhibitors (e.g., donepezil), which are commonly used to prevent memory loss in neurotypical adults with dementia. Actually, adults with DS are at significant risk for the development of AD, with most individuals with DS displaying the neuropathology associated with AD by 40 years of age [11,12]. This is thought to be due in large part to the presence of a third APP gene on the 21st chromosome, resulting in the accelerated production of β-amyloid (referred to as “amyloid”) throughout the lifetime. Amyloid and amyloid plaques are believed to be key to initiating a cascade of subsequent events, including the hyperphosphorylation and accumulation of neurofibrillary tangles comprised of tau protein, as well as changes in brain structure and functioning (e.g., decreased grey matter density, decreased hippocampal volume, increased white matter hyperintensities) that lead to dementia. 

Blood based biomarkers of amyloid peptides (amyloid beta [Aβ 40, 42]) have also been increasingly explored in adults with DS due in part to the early accumulation of this protein. Findings among those with DS and AD have been mixed, with some studies reporting elevations in Aβ 1–42 [13,14,15,16] and Aβ 1–40 [13,14,17,18] while others reporting a relative decrease [15,17,18] in levels, which corresponds with CSF findings [19,20]. Lower levels have also been noted among prodromal AD groups in comparison to healthy controls [13]. In contrast, other plasma biomarkers of AD pathology, including tauopathy (total tau) and neurodegeneration (neurofilament light chain [NfL]), have shown more consistent findings, with elevations seen among individuals with DS who have been diagnosed with AD (DS-AD) [13,15,21,22,23]. 

Finally, there is evidence of even earlier pathological changes in DS, such as increased levels of non-fibrillated amyloid in the teenage years along with non-developmental grey matter and ventricular changes [24]. However, while the symptoms of acute regression appear to mirror some of those of dementia, they differ in that they tend to be sudden rather than gradual and also are not followed by a subsequent and continued loss of skills over a 2–5 year period. Yet, it is also possible that some of the early pathological changes that are documented in adolescents and adults with DS who experience regression, might continue to play a significant role in determining an increased risk for dementia in adulthood.

The Alzheimer’s Biomarker Consortium–Down Syndrome (ABC–DS) is a longitudinal study of risk factors for AD in a large cohort of adults with DS. Funded by the NIA and NICHD, the consortium has enrolled approximately 400 individuals with DS, many of whom have been followed for a number of years. A wide range of potential biomarkers of AD is collected as part of the protocol, including blood and CSF-based measures, and cognitive and adaptive functioning measures, as well as MRI and PET scans. Among the current ABC-DS cohort, five individuals who had a prior documented history of acute regression were identified. By matching them with a group of adults with DS who had not had this experience, we have a unique opportunity to examine a larger number of potential biomarker differences. Drawing on potential causes of acute regression and its possible relation to early dementia, it was hypothesized that those individuals with a history of acute regression would have an increased prevalence of risk biomarkers for AD, including greater levels of amyloid deposition, tau and brain neuropathology, and blood-based biomarkers (e.g., neurofilament light chain [NfL]) than a matched group of unaffected adults. 

## 2. Materials and Methods

### 2.1. ABC-DS 

The ABC-DS comprises eight university-based clinical performance sites, including the University of Pittsburgh, University of Cambridge, Harvard University, Columbia University/IBRDD, Barrow Neurological Institute, University of Wisconsin Madison, University of California Irvine and Washington University. Other sites provide additional support to the project, including the University of North Texas Health Science Center, the University of Southern California, Georgetown University, the University of Michigan and the Mayo Clinic. Study participants undergo a baseline and subsequent follow-up visits at 16-month intervals, for a total of three visits over a 32-month period. 

### 2.2. Participants 

Informed consent/assent was obtained from all participants involved in the study. The study was conducted in accordance with the Declaration of Helsinki, and the protocol was approved by the ethics committees from each of the participating universities. Medical histories of enrollees were reviewed at each clinical performance site to identify those with a history of acute regression. A total of five individuals were identified and medical histories were reviewed from the period of time during which regression purportedly occurred. A diagnosis was confirmed if there was evidence of a significant loss of communication skills and adaptive functioning. As described in Table 1, diagnoses at the time included depression (*N* = 1) and dementia (*N* = 3). Age at the time of reported regression ranged from the early to late 20s. None of the individuals returned to prior levels of functioning. Current estimated mental ages ranged from <2 years 0 months to 6 years 5 months, based upon the PPVT4 [25], a measure of receptive vocabulary. Those initially diagnosed with dementia have since had their diagnoses removed. 

The five individuals with histories of regression were matched with 15 other ABC-DS participants (without reported histories of acute regression) based on biological sex, age, site and ApoE status. None of the individuals with regression, or the matched participants, had current diagnoses of AD based upon a consensus conference (comprised of study staff, a physician and a psychologist who had participated in the study visit) that included a review of neuropsychological assessment battery results, caregiver-completed questionnaires on adaptive functioning, behavioral concerns and possible symptoms of AD. In addition, the consensus conference members had access to each participant’s medical history and the results of a physical/neurological examination conducted as part of their study visit. Consensus conference members were blinded as to neuroimaging, omics and genetics results. Some individuals were given diagnoses of “unable to determine” due to the likely possibility that other factors (e.g., a recent illness, change in living situation or job) might have accounted for any reported changes in overall functioning. ApoE carrier status was obtained and karyotyping was conducted to confirm the trisomy 21 diagnosis. Table 2 provides demographic information for the “regression group” and the 15 “matched controls”. The only meaningful difference between the two groups was estimated mental age, with the regression group having a mean MA considerably lower than the comparison group. 

### 2.3. Dependent Measures 

Methods describing the neuroimaging and omics analyses have been published previously [26]. Structural MRI scans were obtained for all participants using a 3T MRI system with T1-weighted pulse sequences. The following scanners were used: GE Discovery MR750, Siemens Prisma and GE Signa PET/MR. The structural MRI was processed with Freesurfer (version 5.3) to determine hippocampal, caudate and putamen volumes. Both the Aβ and tau PET scans were conducted with all participants, using nominal injections of 15 mCi of [^11^C]PiB and 10 mCi of [^18^F]AV-1451 which were administered as 20–30 s bolus injections followed by a saline flush (one site used lower doses of both imaging agents). Tracer concentration images were generated from 50–70 min post injection for [^11^C]PiB and 80–100 min post injection for [^18^F]AV-1451. Each subject’s PET images were registered to the corresponding T1 image using PMOD.

To provide an amyloid index that could be compared with other populations (e.g., late-onset Alzheimer’s disease, early-onset autosomal dominant Alzheimer’s disease), the [^11^C]PiB uptake was quantified on a universal centiloid scale using the procedure described by Klunk et al. [27]. Briefly, subject T1 MR images along with the [^11^C]PiB images were warped to the Montreal Neurological Institute (MNI)-152 T1-weighted template using SPM8. The warped [^11^C]PiB images were sampled using the centiloid cortex volume of interest (CTX VOI) and normalized using activity sampled with the centiloid whole cerebellum reference region volume of interest (WC VOI). Both VOIs are available at Global Alzheimer’s Association Information Network (GAAIN; http://www.gaain.org (accessed on 22 January 2018)). The determined SUVR was then converted to a centiloid value using equation Equation 1.3b of [27]. Individuals with a centiloid value greater than 22 were considered to be “amyloid positive” [28].

For evaluation of AV-1451 uptake, a probability template method [29] was employed. In this procedure, atlases are defined for a number of template images which are then warped on to the subject’s MR. In the current study, MR images from twelve individuals were used as templates/atlases. The template MRI images were selected through a review of images that had been processed through FreeSurfer 5.3, software which automatically parcellates brain MR images and produces a native space version of the Desikan–Killiany (DK) atlas [30]. Selection of the 12 images for use as templates was based on the quality of the parcellation results. For each participant in this study all 12 template images and corresponding atlases were warped to the participant’s T1 MR, resulting in 12 versions of each DK region. Each final DK region for the participant was taken to be the volume of maximum overlap of the 12. The final DK regions were used to construct the six Braak [31] regions described in Schöll et al. [32] (with the exception that the striatum was not included in the Braak region 5). The uptake of AV-1451 in each of the Braak regions was quantified as regional SUVR, i.e., the activity concentration in the region normalized to cerebellar gray matter activity concentration.

Blood samples were obtained concurrently with the PET scans. Plasma samples were assessed at the University of North Texas Health Science Center (UNTHSC) Institute for Translational Research (ITR) Biomarker Core using a single molecule array technology (Simoa; Quanterix, Billerica, MA, USA). Commercially available kits from Quanterix were utilized. Samples were loaded onto a 96-well plate and analyzed on the Simoa HD-1. Plasma concentrations of Aβ 1–42 and Aβ 1–40, total tau and NfL were obtained for most participants. The UNTHSC ITR Biomarker Core has assayed >5000 on the Simoa platform with coefficients of variability (CVs) <4%. Lower Aβ 1–42 and Aβ 1–40 values and higher total tau and NfL values would be predicted when comparing individuals with regression and unaffected controls.

Statistical Analysis: Due to the small number of individuals identified with histories of regression, only descriptive statistics are provided: means and standard deviations for the continuous measures and counts/frequencies for the dichotomous or non-interval type. A 1:1 and a 1:3 matching were performed based on several variables (age, ApoE status, karyotype and gender). 

## 3. Results

The five participants with histories of regression were matched with 15 unaffected adults with DS. There did not appear to be any meaningful differences between the groups on demographic variables of interest, with the exception of estimated mental age. The MRI for one member of the regression group was unable to be interpreted, due to extensive movement. As a result, data were not available for any of the MRI variables for this individual. The results of the probability template method produced unsatisfactory results for one participant, with the consequence that no tau results were available. As shown in Table 3, there does not appear to be a clinically meaningful difference between groups on mean hippocampal thickness (the two groups differ by less than 2%). Conversely, the regression group actually has slightly larger right and left mean hippocampal, caudate and putamen volumes than the unaffected DS controls. Mean global centiloid SUVR (a measure of brain amyloid) appears to be slightly lower in the regression group than the unaffected controls. However, mean tau PET SUVr units across all six Braak regions suggest minimal differences between the regression group and the 15-member unaffected group. 

Potentially meaningful differences between groups may have been found on some of the proteomics measures (see Table 3). The mean values of both Aβ40 and Aβ42 were slightly lower for those with histories of regression versus unaffected controls (8.7% and 4.3% lower, respectively). As a result, the mean Aβ40/Aβ42 ratio was similarly impacted (with the regression group mean ratio being 4.9% lower than the unaffected control group mean ratio). Potentially meaningful differences were also noted on for both total tau and NfL. The regression group mean NfL value was 37.0% higher than the unaffected controls’ mean NfL values. Similarly, the regression group mean total tau value was 39.3% higher than the unaffected controls’ mean total tau values.

## 4. Discussion

This study sought to examine the possibility that early regression in adults with DS might lead to an increased risk for subsequent AD in later life, resulting in AD symptoms occurring at an earlier age than among unaffected adults with DS. Using data from the ABC-DS study, we were able to identify five individuals with histories of regression during their early to late 20s and to match them with 15 unaffected individuals with DS. While none of the individuals are yet displaying signs of AD, the ABC-DS database allows us to examine the possibility that this group of individuals, with a prior history of regression, might be at increased AD risk based upon a range of AD biomarkers. Hence it was hypothesized that in comparison to matched, unaffected individuals, those with histories of regression would have increased biomarker risk measures for AD, including greater levels of amyloid deposition, tau and brain neuropathology, and blood-based biomarkers (e.g., NfL). While a small N only allowed for the presentation of descriptive statistics, results suggest some clinically meaningful differences between the two groups that could provide preliminary evidence to support this hypothesis.

MRI findings were, in fact, the opposite of what had been hypothesized, with mean hippocampal caudate and putamen volumes being slightly higher in the regression group. Prior research in the adult DS population indicates that hippocampal volume is smaller than in cognitively normal adults and that changes in hippocampal volume among adults with DS are not likely to be seen until after the age of 50 (which also is when many individuals develop dementia) [33]. Similarly, our own prior research found no significant differences in right and left hippocampal volumes when comparing individuals with DS who were amyloid positive versus those who were amyloid negative (neither group had dementia) [34]. Hence, the differences found between the two groups in the current study are likely inconsequential. However, it should also be noted that cortical thickness may be a proxy measure of inflammation and hippocampal volume may be mirroring that. In DS and in autosomal dominant AD, the cortex is at first thicker in areas which are typically affected by AD and these regions then atrophy as disease progresses [35,36,37].

While no differences on tau PET findings were noted, this may have been expected given the mean age of the two groups (38 years). One might not have anticipated high tau PET SUVr values until the middle to late 40s in the DS population. However, there was a difference found on our measure of brain amyloid, with the unaffected group having a 14% greater mean global centiloid SUVR than the regression group (suggesting greater amyloid burden in the former group). This finding was counter to what had been hypothesized. To provide some context for determining if such a difference might have possible clinical significance, we examined findings from our prior research comparing amyloid positive versus amyloid negative adults with DS. In a 2017 paper examining change in amyloid load in a cohort of 52 non-demented adults with DS, those who were amyloid positive (determined via PET scan) had a mean SUVR that was 32.8% higher than those who were amyloid negative [38]. Hence, a 14% difference on mean centiloid SUVR likely has little or no clinical significance in this case. 

Some potential group differences were noted on the proteomics measures. For example, both plasma Aβ40 and Aβ42 were slightly lower in the regression group, which is consistent with prior plasma (15, 17–18) and CSF findings (19–20) that show declines among those with DS-AD, reflecting a similar change among those who regressed. In addition, similar to prior work (13, 15, 21–23), we found considerably higher levels of total tau and NfL, reflecting increased neurodegeneration and further AD specific pathological changes among this group. We compared these findings with those of other ABC-DS papers which examined proteomics differences between individuals who were clinically stable (CS) and those determined to have MCI or AD (based upon a consensus conference decision). In a paper on proteomic profiles in adults with DS, Petersen et al. [39] noted a 6.8% difference between the CS and MCI groups and a 7.6% difference between the CS and AD groups on mean Aβ40. A 3.9% difference between the CS and MCI groups and a 9.1% difference between the CS and AD groups were reported for mean Aβ42 (with Aβ40 and Aβ42 values being lower for the MCI and AD groups). Hence, the mean differences observed between our regression group and our unaffected control group is in this same general range. In a second paper by Petersen et al. [22], a difference of approximately 12% between the CS and MCI groups on mean NfL and a 25% difference between the CS and AD groups on mean total tau was found. It should be noted that in both papers, the MCI and AD groups were significantly older than the CS group (and that many AD biomarkers in adults with DS appear to be significantly impacted by age). As the current regression group and unaffected control group did not differ significantly in age this cannot account for these findings. Hence, the findings suggest greater proteomics biomarker risk in the regression group, reflecting increased AD pathology of amyloid deposition, tau and neurodegeneration. It is possible that, similar to changes in CSF measures of Aβ40, Aβ42, NfL and total tau occur prior to actual detectable changes on amyloid and/or tau PET scans. 

As far as we are aware, this study is the first to make use of amyloid/tau PET scans and proteomics in the examination of individuals with DS who have experienced acute regression. In contrast to prior studies that have reported some MRI results, we were able to include a comparison set of scans for a matched group of unaffected individuals. However, there are a number of weaknesses in this report. First, the N is too small to conduct an adequate statistical analysis of the data. Second, some of the affected individuals did not appear to have been given an official diagnosis of regression at the time of the event (although the description of their behavior is consistent with DS regression). Third, it is possible that this group of five individuals is not representative of the larger group of adolescents and young adults with DS who experience regression. The individuals in this report may have had a more successful recovery, which may have also impacted their risk for AD. Conversely, adults with DS and regression who experienced minimal or no recovery may represent a subgroup that is at the greatest risk for AD. However, such individuals would not have been enrolled in ABC-DS, as a minimum mental age of 36 months was required. It should also be noted that the five individuals with regression were functioning at a much lower level than the comparison group. In fact, it was not possible to create a matched group that had a similar mental age to those who had experienced regression. As a result, we chose not to compare the groups on other cognitive measures, as the possibility could not be excluded that any differences might reflect their level of functioning rather than regression. However, there is no evidence that cognitive level increases the risk for AD among individuals with DS. Therefore, it was not felt that the differences in mental age would be expected to impact the values of our biomarkers of interest.

Finally, while none of these five individuals has yet developed AD (and, in general, are below the age when one would expect to see early AD signs), it is still possible that those who have experienced regression could be more susceptible than nonaffected individuals to experiencing clinically significant changes in AD biomarkers (especially neuroimaging biomarkers) and to exhibiting earlier AD symptoms. Future research should include a larger cohort of individuals who have experienced regression (and include those functioning below a mental age of three years). In addition, other potential proteomics biomarkers, such as ptau181 and ptau217, should be included. Finally, it may require a number of years of follow-up before the potential relationship between regression and increased AD risk is clearer. 

## Figures and Tables

**Table 1 brainsci-11-01109-t001:** Description of participants.

Part	Sex	Current Age	Current IQ/MA	ApoE Status	Age at Regression	Description of Regression
01	M	33	6 years 5 months	E3/E3	28	Individual had some loss of functioning, primarily deterioration in verbal skills and increased irritability. They were given a diagnosis of early dementia at the time. Some improvement occurred over the following 2 years, although not to previous levels of functioning. Dementia diagnosis was subsequently removed.
02	F	44	4 years 0 months	E3/E3	27–28	Individual used to have a part-time job and spoke fluently. After regression, was unable to perform her job and no longer traveled independently. b They also lost interest in most activities and stopped keeping up with friends. No diagnosis of dementia was made. There has been gradual improvement since that time, although not to previous levels of functioning.
03	F	36	3 years 8 months	E3/E3	Mid 20s	Individual was given a dementia diagnosis approximately 10 years ago following a rapid decline in functioning. They were prescribed donepezil and improved a little. The dementia diagnosis was subsequently removed. They continued to improve after medication was discontinued but did not return to prior levels of functioning.
04	M	54	3 years 1 mon.	E4/E3	Early 20s.	Individual had a significant loss of functioning in their early 20s. Prior to regression they were highly verbal and independent. Subsequently, they were unable to talk, complete basic self-care activities, or participate in work and social activities. They were diagnosed with depression and prescribed an SSRI. There was some minimal improvement in language subsequently noted along with some improvement in self-care and social skills. They have maintained this same level of functioning since then and continue to have a diagnosis of depression and to take medication. They were never officially diagnosed with regression.
05	F	28	<2 years 0 months	E3/E3	Mid 20s	Individual evidenced significant loss of skills in early 20s and was diagnosed with dementia. Some small gains have been noted since that time, but not to prior levels of functioning. The dementia diagnosis was removed, however, significant behavioral issues continue along with limited language.

**Table 2 brainsci-11-01109-t002:** Characteristics of regression and matched groups.

Variable	Regression Group	Matched Group 1:15
Age [Mean (SD)/N(%)]	38.06 (8.78)	38.01 (8.06)
Mental Age	3.42 (2.30)	8.16 (4.40)
Sex (Male)	2 (40%)	6 (40%)
ApoE Allele *	1 (20%)	3 (20%)
Karyotype		
Full Trisomy 21	4 (80%)	12 (80%)
Partial Trisomy	1 (20%)	3 (20%)
Cognitive Level		
Mild	0 (0%)	4 (26.7%)
Moderate	3 (60%)	11 (73.3%)
Severe	2(40%)	0 (0%)
Consensus Diagnosis		
Cognitively Stable	4 (80.0%)	13 (86.7%)
Unable to Determine	1 (20.0%)	2 (13.3%)

* ApoE4 = apolipoprotein E gene variant 4.

**Table 3 brainsci-11-01109-t003:** Biomarkers.

	Regress. Group (N)	Mean (SD)	Median	Matched Group 1:15 (N)	Mean (SD)	Median
**MRI Scan**						
**Hippocampal Volume**						
Left	4	3625.3 (283.8)	3618.4	11	3174.7 (435.2)	3163.2
Right	4	3655.1 (299.2)	3653.3	11	3156.3 (498.0)	3120.9
**Hippocampal Thickness**	4	3.00 (0.11)	2.98	11	2.95 (0.21)	2.86
**Caudate**						
Left	4	3471.6 (415.4)	3372.6	11	3219.0 (579.6)	3256.4
Right	4	3505.3 (475.5)	3447.4	11	3310.7 (375.8)	3352.0
**Putamen**						
Left	4	6205.7 (453.7)	6375.3	11	5408.4 (748.4)	5149.4
Right	4	6031.2 (356.6)	5903.3	11	5523.5 (578.2)	5300.0
**Amyloid PET**						
Centiloid SUVR	5	17.77 (22.09)	6.37	15	20.70 (22.59)	18.66
Amyloid Negative ≤ 22		3 (60%)			9 (60%)	
Amyloid Positive >= 22		2 (40%)			6 (40%)	
**TAU PET**						
Braak 1	4	1.22 (0.16)	1.20	15	1.19 (0.19)	1.14
Braak 2	4	1.17 (0.26)	1.12	15	1.18 (0.17)	1.11
Braak 3	4	1.11 (0.09)	1.09	15	1.13 (0.14)	1.09
Braak 4	4	1.08 (0.08)	1.10	15	1.11 (0.12)	1.08
Braak 5	4	1.08 (0.04)	1.10	15	1.09 (0.13)	1.07
Braak 6	4	1.06 (0.04)	1.08	15	1.03 (0.07)	1.02
**Plasma**						
Aβ40 pg/mL	4	413.25 (52.51)	403	14	452.43 (86.48)	452
Aβ42 pg/mL	4	15.13 (2.89)	14.60	14	15.81 (3.37)	15.60
Aβ40/ Aβ42 ratio	4	27.64 (2.93)	27.43	14	29.06 (4.34)	29.25
NfL pg/mL	5	18.23 (11.37)	15.20	15	11.49 (6.51)	10.80
Total Tau pg/mL	4	6.13 (6.55)	3.26	14	3.72 (4.09)	2.68

## Data Availability

Requests for qualified investigators to obtain data supporting the reported results can be made at https://pitt.co1.qualtrics.com/jfe/form/SV_cu0pNCZZlrdSxUN, accessed on 13 August 2021.

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
