# Peer review of "Acute Regression in Down Syndrome"

_brainsci, 2021, doi:10.3390/brainsci11081109_

Round 1
Reviewer 1 Report
The issue of regression in Down syndrome is poorly understood and, as a result, poorly treated. This paper explores the potential link between regression and Alzheimer's disease by looking at AD biomarkers. While the number of participants was low making it difficult to draw many conclusions, the importance of the research make it important to share the data with Down syndrome researchers. As such, I strongly support the publication of this paper. There are just a few minor edits. On line 48, the word "of" should be removed. On line 60, there is a typo with the word "results". On line 80, I suggest mentioning amyloid and amyloid plaques and replacing "the protein, tau" with "neurofibrillary tangles comprised of tau protein" as this is the toxic form of tau. The authors should also take care to spell out all abbreviations the first time that they are used.
Author Response
Thank you so much for the positive review and feedback. My responses follow.
- On line 48, the word "of" should be removed. Response: Done
- On line 60, there is a typo with the word "results". Response: Done
- On line 80, I suggest mentioning amyloid and amyloid plaques and replacing "the protein, tau" with "neurofibrillary tangles comprised of tau protein" as this is the toxic form of tau. Response: Done
- The authors should also take care to spell out all abbreviations the first time that they are used. Response: Have spelled out all abbreviations the first time they are used.
Reviewer 2 Report
Handen et al. do an exploratory description of the differences in neuroimaging and plasma proteins between DS subjects who have suffered acute regression and control DS subjects. Although the sample size does not allow them to do statistical analysis, they do a thorough comparison of both groups. Their main hypothesis is that those that have suffered acute regression will have measures that resemble DS subjects with Alzheimer’s disease. However, this hypothesis proves to not be valid for the study population. As they explain, it might be because the included subjects presented an improvement, and those that did not recover and went on to develop Alzheimer’s were excluded from the study (due to the study inclusion criteria).
The paper is generally well written and is of interest, nevertheless, it could benefit from some changes or additions to better discuss the biomarkers and their relationship to Alzheimer’s disease in DS.
One of my main concerns is the use of the term “omics” throughout the paper, although only a targeted immuno-based protein analysis was performed (and on platforms that are not usually considered proteomics). The authors may consider clarifying this as the reader could expect more data (including other "omics").
Introduction:
On lines 79-80, the authors state that “Amyloid is thought to be key to initiating a cascade of subsequent events, including the appearance of the protein, tau, as well” this sentence is unclear and misleading. Tau, as a protein, is always present, however, Abeta peptides can lead to the hyperphosphorylation and the accumulation of neurofibrillary tangles. Some papers that show it (non-exhaustive list):
- Oliveira JM, Henriques AG, Martins F, Rebelo S, da Cruz e Silva OA. Amyloid-β Modulates Both AβPP and Tau Phosphorylation. J Alzheimers Dis. 2015;45(2):495-507. doi: 10.3233/JAD-142664. PMID: 25589714.
- Zheng WH, Bastianetto S, Mennicken F, Ma W, Kar S. Amyloid beta peptide induces tau phosphorylation and loss of cholinergic neurons in rat primary septal cultures. 2002;115(1):201-11. doi: 10.1016/s0306-4522(02)00404-9. PMID: 12401334.
- Hsin-Yi Wu, PhD, Po-Cheng Kuo, MS, Yi-Ting Wang, PhD, Hao-Tai Lin, MS, Allyson D Roe, BS, Bo Y Wang, MS, Chia-Li Han, PhD, Bradley T Hyman, MD, PhD, Yu-Ju Chen, PhD, Hwan-Ching Tai, PhD, β-Amyloid Induces Pathology-Related Patterns of Tau Hyperphosphorylation at Synaptic Terminals, Journal of Neuropathology & Experimental Neurology, Volume 77, Issue 9, September 2018, Pages 814–826, https://doi.org/10.1093/jnen/nly059
I am not aware of the word limit for the introduction, but I feel that a small introduction on the plasma biomarkers would be beneficial. Specially talking about the specificity to detect AD or prodromal changes of each of them.
Methods:
Why was phospho-tau not measured? The relationship of the different species of phosphorylated tau to Alzheimer is different and has been extensively documented (some examples (non-exhaustive)):
- Yang CC, Chiu MJ, Chen TF, Chang HL, Liu BH, Yang SY. Assay of Plasma Phosphorylated Tau Protein (Threonine 181) and Total Tau Protein in Early-Stage Alzheimer's Disease. J Alzheimers Dis. 2018;61(4):1323-1332. doi: 10.3233/JAD-170810. PMID: 29376870.
- Nicolas R. Barthélemy, Kanta Horie, Chihiro Sato, Randall J. Bateman; Blood plasma phosphorylated-tau isoforms track CNS change in Alzheimer’s disease. J Exp Med 2 November 2020; 217 (11): e20200861. doi: https://doi.org/10.1084/jem.20200861
- Thijssen, E.H., La Joie, R., Wolf, A. et al. Diagnostic value of plasma phosphorylated tau181 in Alzheimer’s disease and frontotemporal lobar degeneration. Nat Med 26, 387–397 (2020). https://doi.org/10.1038/s41591-020-0762-2
Results:
Table 3: Considering the small number of subjects and the fact that the SD for the values are so different between groups, the authors might consider adding the median values. The addition of those values could paint a better picture of the real differences between the two groups.
Discussion:
Considering the results were lower SUVr and plasma amyloid was observed, can the authors really conclude that the affected subjects have higher AD biomarkers? (Lines 238-244).
The part of the discussion regarding plasma biomarkers could benefit from a rewrite. The authors have clumped together the different biomarkers without taking into account the different aspects of neurodegeneration that they represent. NfL is known to be a biomarker of neurodegeneration independently of cause, as is total tau. The authors already cite the work by Fortea et al. but they do not include it on the discussion with relation to biomarkers. Other papers by the same group could also clarify these aspects.
- Fortea J, Carmona-Iragui M, Benejam B, Fernández S, Videla L, Barroeta I, Alcolea D, Pegueroles J, Muñoz L, Belbin O, de Leon MJ, Maceski AM, Hirtz C, Clarimón J, Videla S, Delaby C, Lehmann S, Blesa R, Lleó A. Plasma and CSF biomarkers for the diagnosis of Alzheimer's disease in adults with Down syndrome: a cross-sectional study. Lancet Neurol. 2018 Oct;17(10):860-869. doi: 10.1016/S1474-4422(18)30285-0. Epub 2018 Aug 29. PMID: 30172624.
What are the true meanings of increased NfL and tau? The ratio Ab40/Ab42 suggests a lower Ab40 production compared to Ab42, or considering it from another point of view, a higher production of the longer and more toxic form. As said precedently, NfL is a marker of neurodegeneration, however, the cortical thickness and the volumes, as seen on MRI, are increased in the affected group. The authors suggest the possibility of neuroinflammation, could they link it to the plasma biomarkers?
The authors discuss the fact that subjects that have had a less successful recovery, may not have been included in this study because of the inclusion criteria. However, throughout the paper, they underline the fact that the subjects in the affected group have a lower mental age. Are subjects with a lower mental age more likely to present acute regression?
Which are the future directions that the authors would consider to better study this phenomena in DS?
Minor comments:
The authors might consider the use of “β-amyloid” throughout the text instead of only “amyloid”, as Aβ is not the only peptide that produces amyloid plaques (in their original sense). However, I am aware that the context does help clarify the amyloid the authors are refereeing to.
Line 60: "rresults"
Line 100: "Drawng"
Line 133: "biolgical"
Author Response
Response to Reviewer 2
We would like to thank Reviewer 2 for their detailed feedback and suggestions. Our responses are below:
Introduction:
On lines 79-80, the authors state that “Amyloid is thought to be key to initiating a cascade of subsequent events, including the appearance of the protein, tau, as well” this sentence is unclear and misleading. Tau, as a protein, is always present, however, Abeta peptides can lead to the hyperphosphorylation and the accumulation of neurofibrillary tangles. Some papers that show it (non-exhaustive list):
Response: We have reworded this section based upon both reviewers’ feedback.
I am not aware of the word limit for the introduction, but I feel that a small introduction on the plasma biomarkers would be beneficial. Specially talking about the specificity to detect AD or prodromal changes of each of them.
Response: We have added the following paragraph to the introduction with appropriate citations.
Blood based biomarkers of amyloid peptides (amyloid beta [Aβ40, 42]) have been increasingly explored in adults with DS due in part to the early accumulation of this protein. Findings among those with DS and Alzheimer’s disease (AD) have been mixed though with some studies findings elevations in Aβ1-42 (ref) and Aβ 1-40 (ref) while other find a relative decrease (ref) in levels, which corresponds with CSF findings (ref). Lower levels were also noted for prodromal AD groups as compared to health controls (ref). In contrast, other plasma biomarkers of AD pathology including tauopathy (total tau) and neurodegeneration (neurofilament light chain [NfL]) have shown more consistent findings with elevations seen among those with DS-AD (ref).
Methods:
Why was phospho-tau not measured? The relationship of the different species of phosphorylated tau to Alzheimer is different and has been extensively documented (some examples (non-exhaustive)):
Response: P-tau (181 or 217) data was not yet available for the ABC-DS cohort. The assays were conducted for this study using a 3-plex assay plate, which included amyloid beta (Aβ 40, Aβ 42) and total tau and, therefore, these specific proteins were available for review. Additional work is ongoing including collaborations to generate p-tau data on this very unique cohort so that future work might be able to further analyze this. We have included the recommendation to add ptau181 and 217 to future research.
Results:
Table 3: Considering the small number of subjects and the fact that the SD for the values are so different between groups, the authors might consider adding the median values. The addition of those values could paint a better picture of the real differences between the two groups.
Response: Median values have been added to table 3.
Discussion:
Considering the results were lower SUVr and plasma amyloid was observed, can the authors really conclude that the affected subjects have higher AD biomarkers? (Lines 238-244).
Response: We did not believe that we concluded that the affected subjects had higher AD biomarkers. We indicated that some clinically meaningful differences between the two groups were noted among the proteomics biomarkers (but not the MRI or PET findings).
The part of the discussion regarding plasma biomarkers could benefit from a rewrite. The authors have clumped together the different biomarkers without taking into account the different aspects of neurodegeneration that they represent. NfL is known to be a biomarker of neurodegeneration independently of cause, as is total tau. The authors already cite the work by Fortea et al. but they do not include it on the discussion with relation to biomarkers. Other papers by the same group could also clarify these aspects.
Response: We have revised this section based upon the reviewer’s suggestions.
What are the true meanings of increased NfL and tau? The ratio Ab40/Ab42 suggests a lower Ab40 production compared to Ab42, or considering it from another point of view, a higher production of the longer and more toxic form. As said precedently, NfL is a marker of neurodegeneration, however, the cortical thickness and the volumes, as seen on MRI, are increased in the affected group. The authors suggest the possibility of neuroinflammation, could they link it to the plasma biomarkers?
Response: We have added to this to the appropriate section in the Discussion.
The authors discuss the fact that subjects that have had a less successful recovery, may not have been included in this study because of the inclusion criteria. However, throughout the paper, they underline the fact that the subjects in the affected group have a lower mental age. Are subjects with a lower mental age more likely to present acute regression?
Response: There is no data to suggest that individuals with DS who have a lower mental age are more likely to experience regression. However, following a period of regression, many individuals fail to return to pre-regression levels of functioning. As we discuss in the Introduction, many never recover language or basic life skills. Individuals at this level of functioning would not have met inclusion criteria for the ABC-DS study.
Which are the future directions that the authors would consider to better study this phenomena in DS?
Response: We have included suggestions for future directions at the end, including having a larger cohort, following individuals for a more extended period, and the inclusion of other biomarkers such as ptau181 and 217.
Minor comments:
The authors might consider the use of “β-amyloid” throughout the text instead of only “amyloid”, as Aβ is not the only peptide that produces amyloid plaques (in their original sense). However, I am aware that the context does help clarify the amyloid the authors are refereeing to.
Response: We agree that the use of the term Aβ is more accurate. However, most of our previous papers have used the term “amyloid” throughout and we feel that this continues to be appropriate. We have, instead, used the term “β-amyloid” when first mentioned and indicated that it will be referred to as “amyloid” in the rest of the paper.
Line 60: "rresults" Done. Thank you.
Line 100: "Drawng" Done. Thank you.
Line 133: "biolgical" Done. Thank you.